# Linkage to primary care after home-based blood pressure screening in rural KwaZulu-Natal, South Africa: a population-based cohort study

Mark J Siedner,[1,2,3] Kathy Baisley,[3,4] Joanna Orne-Gliemann,[5] Deenan Pillay,[3,6] Olivier Koole,[3,4] Emily B Wong,[1,2,3] Philippa Matthews,[3] Frank Tanser,[3] Kobus Herbst,[3] Till Barnighausen,[3,6,7,8] Max Bachmann[9]

For numbered affiliations see end of article.

**Correspondence to**
Dr Mark J Siedner;
msiedner@mgh.harvard.edu

## ABSTRACT

**Objectives** The expanding burden of non-communicable diseases (NCDs) globally will require novel public health strategies. Community-based screening has been promoted to augment efficiency of diagnostic services, but few data are available on the downstream impact of such programmes. We sought to assess the impact of a home-based blood pressure screening programme on linkage to hypertension care in rural South Africa.

**Setting** We conducted home-based blood pressure screening withinin a population cohort in rural KwaZulu-Natal, using the WHO Stepwise Approach to Surveillance (STEPS) protocol.

**Participants** Individuals meeting criteria for raised blood pressure (≥140 systolic or ≥90 diastolic averaged over two readings) were referred to local health clinics and included in this analysis. We defined linkage to care based on self-report of presentation to clinic for hypertension during the next 2 years of cohort observation. We estimated the population proportion of successful linkage to care with inverse probability sampling weights, and fit multivariable logistic regression models to identify predictors of linkage following a positive hypertension screen.

**Results** Of 11 694 individuals screened, 14.6% (n=1706) were newly diagnosed with elevated pressure. 26.9% (95% CI 24.5% to 29.4%) of those sought hypertension care in the following 2 years, and 38.1% (95% CI 35.6% to 40.7%) did so within 5 years. Women (adjusted OR (aOR) 2.41, 95% CI 1.68 to 3.45), those of older age (aOR 11.49, 95% CI 5.87 to 22.46, for 45–59 years vs <30) and those unemployed (aOR 1.71, 95% CI 1.10 to 2.65) were more likely to have linked to care.

**Conclusions** Linkage to care after home-based identification of elevated blood pressure was rare in rural South Africa, particularly among younger individuals, men and the employed. Improved understanding of barriers and facilitators to NCD care is needed to enhance the effectiveness of blood pressure screening in the region.

## BACKGROUND

Over two in three deaths worldwide are attributed to non-communicable diseases (NCDs).[1] Although precise measurement

### Strengths and limitations of this study

► Applies a longitudinal population cohort study design with a large sample size to assess linkage to hypertension care after a home-based screening for elevated blood pressure.
► Assesses a population in rural sub-Saharan Africa who are noted to have high prevalence of hypertension but with little corresponding data about linkage to care after diagnosis.
► Identifies low rates of linkage to care after home-based blood pressure screening in this population, and key factors associated with poor linkage including male sex, younger age and being employed.
► Limitations include low rates of participation in the home-based screening programme and incomplete follow-up, as well as self-reported linkage to care as an outcome definition.

of cause-specific mortality in much of the developing world remains a challenge, some estimates suggest that the majority of NCD deaths now occur in low and middle-income countries.[2] In South Africa, for example, the WHO estimates that half of deaths are due to NCDs, and approximately 25% of the population will suffer a premature death due to them.[3]

Consequently, responding to the NCD epidemic in low and middle-income countries is both a major challenge and stated priority of the public health community.[4] NCD morbidity and mortality can be substantially reduced through effective primary and secondary prevention measures targeting risk factors such as smoking, high blood pressure, diabetes, diet and physical activity.[5] Hypertension, which can be controlled through cost-effective lifestyle and pharmacotherapy interventions, is estimated to account for over 50% of the population attributable fraction

of stroke in the African region.[6] Yet, in South Africa, national population surveys have estimated that over a quarter of South Africans adults have raised blood pressure, but only approximately one in three of them has received treatment.[7]

The South African Department of Health has outlined strategic NCD goals which highlight the role for prevention of NCDs and the importance of a community-based focus.[8] One specific strategy includes integrating HIV and NCD screening programmes and broadening access to diagnostic and treatment in the community and rural areas. Community-based NCD screening through health fairs and use of community health workers has gained traction recently as a means to efficiently screen large populations of individuals for multiple comorbidities.[9 10] Whether such endeavours lead to successful linking of individuals to appropriate NCD care is not well established, and is an important question for the field.

In 2010, we conducted a home-based assessment of blood pressure in approximately 12 000 people in a demographic health surveillance (DHS) site in KwaZulu-Natal. We referred individuals with raised blood pressure and not already receiving hypertension treatment to local government clinics for repeat measurement and ongoing hypertension care. We assessed linkage to care during future years of the home-based DHS survey. Our primary aims were to determine the probability of clinical engagement within 2 years after home-based screening and referral, and to identify predictors of failure to link to care. Our overarching aim was to inform public health programmers on the feasibility of community-based blood pressure screening as an entry point into NCD care in this setting.

## METHODS
### Study design, setting and participants
The African Health Research Institute (formerly the Africa Centre for Health and Population Studies) is a Wellcome Trust funded research institute in South Africa. Since 2000, they have conducted a population cohort study of all adults in a catchment area of 438 km$^2$ in rural uMkhanyakude District, northern KwaZulu-Natal, covering a total population of approximately 100 000 individuals.[11] Households are surveyed 2–3 times per year, to collect information on birth, deaths and migration patterns for all household members, including non-residents. Since 2003, resident household members≥15 years have been invited to participate in an annual home-based individual survey which collects data on sociodemographics and general health information.

### Blood pressure screening and referral methods
In 2010, all individuals who participated in the home-based survey were also offered a physical examination to determine weight, height and blood pressure, using the WHO STEPS protocol.[12] Blood pressure was measured using Omron automated blood pressure

monitors (Omron Global, Kyoto Japan). Blood pressure was measured after 15 min of resting in a seated position. We collected three measurements, each 5 min apart, with the mean of the last two measurements used to identify those with elevated blood pressure. A positive hypertension screen was defined as a mean systolic blood pressure ≥140 mm Hg and/or diastolic blood pressure ≥90 mm Hg.[13] All individuals were given a copy of their blood pressure results, along with body mass index (BMI), on a health results card. Those with elevated blood pressure were counselled by study staff to seek care at their preferred local public health clinic with a copy of their screening results.

### Outcome assessment
For our primary outcome of interest, we defined successful linkage to care for hypertension as self-reported linkage within 2 years of a positive home-based hypertension screen. To assess this outcome, we used data from the two subsequent, annual home-based health surveys in 2011 and 2012. All participants who took part in the initial blood pressure screening survey were members of the demographic health and surveillance site study population, and are seen annually at home for data collection. In each annual health survey, respondents were asked if: (1) they have been diagnosed with hypertension in the past 12 months, (2) if they have ever received hypertension treatment and (3) if they are currently being treated for hypertension. We defined successful linkage to care by a positive response to any of these three questions in either 2011 or 2012. As secondary outcomes of interest, we also examined (1) linkage to hypertension care within 5 years (as opposed to two), defined as a positive response to any of the same three questions in the annual health surveys during 2011–2015 and (2) confirmation of hypertension care 7 years after the screening, as evidenced by clinical records from all public health clinics in the catchment area in 2017, the first year linked clinical data were linked to the population cohort database.

### Statistical methods
We included in this analysis individuals who had a positive hypertension screen in the home-based 2010 survey, and who reported no previous diagnosis of hypertension or hypertension treatment. We first summarised sociodemographic characteristics of eligible adults who had blood pressure measurements in the 2010 survey. We then estimated population-level prevalence of linkage to hypertension care in the 2 years after the screening programme, both overall and stratified by sex and age, with the use of inverse probability weights (IPWs) of the probability of participating in the hypertension screening. We used IPWs to make the results generalisable to the entire 2010 sample. To calculate the weights, we fit a logistic regression model with completion of blood pressure screening in 2010 as the outcome of interest and included age strata, sex, education level and place of residence (urban,

periurban or rural) as predictors, based on information routinely collected in the household-level survey.

We then fit logistic regression models to estimate ORs and 95% CIs for factors independently associated with linkage to hypertension care within 2 years of a positive home-based hypertension screen. Potential determinants of linkage were examined at three levels: sociodemographic factors (age, sex, educational attainment, employment status and socioeconomic status which was estimated using principal components analysis of household asset ownership following the method of Filmer and Pritchett[14]), geographical factors (distance from clinic, urban vs rural residency) and clinical factors (BMI, elevated blood pressure severity (defined using hypertension stages as (a) stage I: systolic 140–160 and diastolic 80–100, (b) stage II: systolic 160-<180 or diastolic 100-<120, or (c) hypertensive urgency: systolic ≥180 or diastolic ≥120), self-report of diabetes, self-report of tuberculosis).[15] Self-reported data on HIV diagnosis were not collected in the survey, so could not be included in the analysis. Sociodemographic and clinical factors whose age- and sex-adjusted association with linkage was significant at p<0.10 were included in a final adjusted multivariable model. Distance from the nearest clinic was analysed as a continuous covariate. In order to allow for non-linear relationships between distance and linkage to care, we used fractional polynomial functions.[16]

We tested the robustness of our findings using several sensitivity analyses. First, we changed our outcome from self-reported linkage to care in 2011 or 2012 to (1) self-reported linkage to care at any time between 2011 and 2015, and (2) confirmation of a clinic appointment for hypertension in 2017 at any of the 11 local public sector clinics, among those who remained a resident in the catchment area. Next, we compared characteristics of eligible individuals who did and did not complete blood pressure screening in 2010. Next, we compared characteristics of those who participated in a subsequent health survey and those who did not, either because of refusal, out-migration or death. Finally, we conducted sensitivity analyses in which we: (1) used IPWs of screening in the models and (2) added a covariate to indicate the number of individual health surveys participated in during 2011–2015. Data were entered and verified in an SQL database and were analysed using Stata V.14 (StataCorp).

## Patient and public involvement

Patients were not involved in the design of this study. This analysis was designed by study investigators at the Africa Health Research Institute intent on leveraging prior home-based screening protocols to inform and optimise future community-based research, and particularly to improve the public health impact of such activities. The results of this study were presented to the South African Department of Health Non-Communicable Diseases Unit and will be disseminated to the community during the monthly scheduled Africa Health Research Institute community road shows.

# RESULTS
## Survey participants

A total of 37 693 potentially eligible adults were in the sampling frame. Of these, approximately one quarter (8589, 22.8%) were not available due to out-migration, death or inability to consent and another 2920 (7.7%) could not be contacted (figure 1). Of the remaining 26 184 individuals who were contacted and eligible for the home-based DHS survey in 2010, 11 814 (45.1%) consented to participate in the general health survey and 11 694 (44.7%) had valid blood pressure measurements. Women, older individuals and those of lower socioeconomic position and education were more likely to participate in the survey (online supplementary table 1).

The majority of participants with a blood pressure measurement were women (n=8241, 70.5%, table 1). Median age was 25 years (IQR 18–47 years) for men and 38 years (IQR 23–55 years) for women. The majority of participants (n=7464, 63.8%) resided in a rural setting, and less than one quarter (n=2642, 22.6%) lived within 1.5 km of the nearest clinic. Few participants (n=1779, 15.2%) were currently employed.

## Screening for hypertension

Approximately one quarter (n=3074, 26.2%) of participants were found to have elevated pressure during the home-based blood pressure screening, of whom 1368 (44.5%) reported having been previously diagnosed or currently on treatment. Of those who had been previously diagnosed or in hypertension care, 1169 (85.5%) were currently on hypertension treatment. Participants who were not previously aware of their condition were significantly younger, and more likely to be men, married, employed, have a higher level of education and be living in periurban areas than those who had been previously diagnosed or on treatment (table 1). However, there was no evidence of a difference between the two groups in the distance from their nearest clinic.

## Analytic sample

A total of 1199 individuals (70.3%) who were not previously aware of having elevated blood pressure participated in a second general health survey within 2 years of being screened (ie, in 2011 or 2012), and were included in the primary analysis of factors associated with linkage to hypertension care. Compared with the 507 individuals who did not participate in 2011 or 2012, those who participated in 2011 or 2012 were older (median (IQR) age=50 (38–66) years, vs 43 (29–58) years), more likely to be women, unmarried, have lower levels of education, be unemployed and have a higher BMI (online supplementary table 2). There was no difference in participation rates by distance from the nearest clinic. When we expanded the observation period to include surveys from 2011 to 2015, a total of 1421 (83.3%) participated in at least one home-based annual general health survey. Of the 285 (16.7%) individuals who did not participate in any health survey after 2010, 81 out-migrated and 36

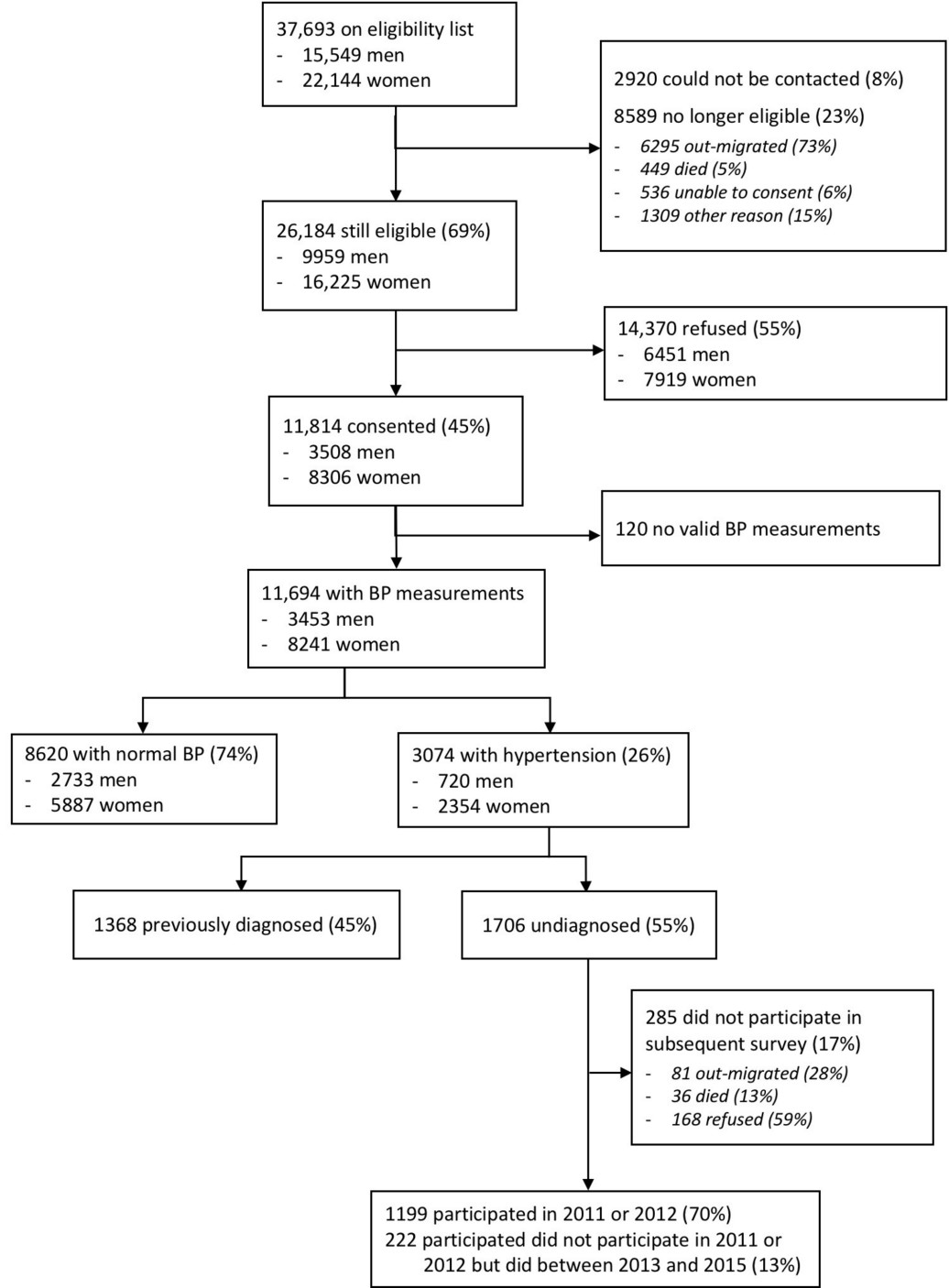

**Figure 1** Flow diagram of eligible and included participants in a baseline community-based hypertension screen in 2010 and follow-up observation during 2011/2012. BP, blood pressure.

died before the 2011 survey (figure 1). The remaining 168 were eligible for at least one subsequent survey but refused participation.

### Linkage to subsequent hypertension care

The crude and, IPW-adjusted population prevalence of linkage to hypertension care within 2 years of the blood pressure screen was 28.6% (95% CI 26.1% to 31.2%) and 26.9% (95% CI 24.5% to 29.4%), respectively. Of the 343 total new linkages reported, 218 (64%) and 135 (36%) were reported in 2011 and 2012, respectively. Of the 218 linkages reported in 2011, the majority (191, 88%) reported initiating antihypertensive therapy. Using IPW-adjusted estimates, we found that women were more likely than men to link to care, and older individuals were more likely than younger individuals (figure 2), such that we estimate that 44.9% (95% CI 39.4% to 50.5%) of women ≥60 years presented to care for hypertension in the next 2 years, versus only 3.0% (95% CI 1.1% to

**Table 1** Participants with blood pressure measured in 2010 survey, those with hypertension* and stratified by whether or not previously diagnosed or on treatment

| | All participants | N with hypertension* | Previously diagnosed† | Not previously diagnosed‡ |
|---|---|---|---|---|
| | n=11694 | n=3074 (26.3%) | n=1368 (44.5%) | n=1706 (55.5%) |
| Median (IQR) age (years) | 34 (21–53) | 55 (43–68) | 60 (51–71) | 49 (35–63) |
| Age group | | | | P<0.001§ |
| <30 | 5107 (43.7%) | 354 (11.5%) | 21 (1.5 %) | 333 (19.5%) |
| 35–44 | 2191 (18.7%) | 484 (15.7%) | 118 (8.6 %) | 366 (21.5%) |
| 45–59 | 2265 (19.4%) | 992 (32.3%) | 507 (37.1%) | 485 (28.4%) |
| 60+ | 2131 (18.2%) | 1244 (40.5%) | 722 (52.8%) | 522 (30.6%) |
| Sex | | | | P<0.001 |
| Male | 3453 (29.5%) | 720 (23.4%) | 178 (13.0%) | 542 (31.8%) |
| Female | 8241 (70.5%) | 2354 (76.6%) | 1190 (87.0%) | 1164 (68.2%) |
| Education | | | | P<0.001 |
| None | 2389 (20.5%) | 1032 (33.6%) | 550 (40.2%) | 482 (28.3%) |
| Less than complete secondary | 6244 (53.5%) | 1463 (47.6%) | 662 (48.4%) | 801 (47.0%) |
| Complete secondary/above | 3040 (26.0%) | 576 (18.8%) | 155 (11.3%) | 421 (24.7%) |
| *Missing* | *21* | *3* | *1* | *2* |
| Marital status | | | | P<0.001 |
| Single (never married) | 3462 (29.8%) | 518 (16.9%) | 174 (12.7%) | 344 (20.2%) |
| Married/informal union | 6556 (56.3%) | 1696 (55.2%) | 688 (50.3%) | 1008 (59.2%) |
| Widowed/separated/divorced | 1618 (13.9%) | 857 (27.9%) | 506 (37.0%) | 351 (20.6%) |
| *Missing* | *58* | *3* | *0* | *3* |
| Employed | | | | P<0.001 |
| Yes | 1779 (15.3%) | 437 (14.2%) | 140 (10.2%) | 297 (17.4%) |
| No | 9828 (84.7%) | 2634 (85.8%) | 1228 (89.8%) | 1406 (82.6%) |
| *Missing* | *87* | *3* | *0* | *3* |
| Residence | | | | P<0.001 |
| Urban | 617 (5.3 %) | 119 (3.9 %) | 55 (4.0 %) | 64 (3.8 %) |
| Periurban | 3604 (30.8%) | 904 (29.4%) | 347 (25.4%) | 557 (32.7%) |
| Rural | 7464 (63.9%) | 2050 (66.7%) | 966 (70.6%) | 1084 (63.6%) |
| *Missing* | *9* | *1* | *0* | *1* |
| SES tertile | | | | P=0.67 |
| Low | 4193 (36.4%) | 1173 (38.6%) | 525 (38.7%) | 648 (38.5%) |
| Middle | 3818 (33.1%) | 947 (31.2%) | 412 (30.4%) | 535 (31.8%) |
| High | 3522 (30.5%) | 918 (30.2%) | 418 (30.8%) | 500 (29.7%) |
| *Missing* | *161* | *36* | *13* | *23* |
| Self-report of diabetes¶ | | | | P<0.001 |
| No | 11 300 (96.6%) | 2867 (93.3%) | 1176 (86.0%) | 1691 (99.1%) |
| Yes | 394 (3.4 %) | 207 (6.7 %) | 192 (14.0%) | 15 (0.9 %) |
| Nearest clinic (km)** | | | | P=0.84 |
| 0–<1.5 | 2642 (22.6%) | 676 (22.0%) | 292 (21.3%) | 384 (22.5%) |
| 1.5–2.5 | 2879 (24.6%) | 710 (23.1%) | 314 (23.0%) | 396 (23.2%) |
| >2.5–3.9 | 2975 (25.5%) | 809 (26.3%) | 368 (26.9%) | 441 (25.9%) |
| >3.9 | 3189 (27.3%) | 878 (28.6%) | 394 (28.8%) | 484 (28.4%) |
| *Missing* | *9* | *1* | *0* | *1* |

Italic values are for missing data, which are typically highlighted as separate (and not analyzed) data.
*Hypertension defined as systolic BP ≥140 mm Hg or diastolic BP ≥90 mm Hg, in an average of two readings.
†Report having been previously diagnosed or on treatment for hypertension in 2010 survey.
‡Report no previous diagnosis or treatment for hypertension in 2010 survey.
§P value from $\chi^2$ test comparing characteristics of those previously diagnosed/treatment and those with no previous diagnosis/treatment.
¶Report having been diagnosed with or on treatment for diabetes in 2010 survey.
**Quartiles based on distribution in all individuals who were eligible for 2010 survey.
BP, blood pressure; SES, socioeconomic status.

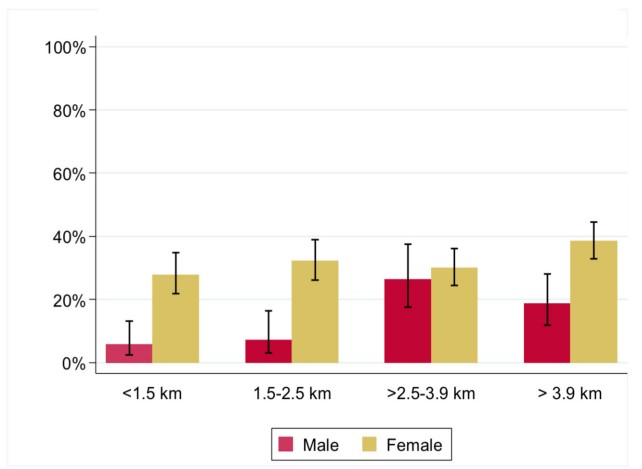

**Figure 2** Proportion of individuals linked to hypertensive care 2 years after a new notification of elevated blood pressure (weighted estimates).

7.7%) of men under 45. When we extended our surveillance period out to 2015, we estimate that 38.1% (95% CI 35.6% to 40.7%) of individuals reported linking to hypertension care within 5 years. Finally, we found that only 16.6% (95% CI 14.6% to 18.9%) of individuals who remained a resident in 2016 and who screened positive for elevated blood pressure in 2010 completed a clinic appointment for hypertension at one of the public health clinics in the catchment area in 2016.

### Factors associated with presentation to hypertension care within 2 years

In models adjusted for socioeconomic, geographic and clinical factors, we found strong evidence that women (OR=2.76, 95% CI 1.97 to 3.88, p<0.001) and those of older age (OR=12.89, 95% CI 6.62 to 25.11, p<0.0001, comparing those 45–59 years vs those <30) were more likely to present to hypertension care within 2 years of home-based diagnosis (table 2). In adjusted analysis, the association with age and sex remained statistically significant, and there was no evidence that the effect of age on linkage to care differed between men and women (p value for interaction=0.20, figure 2). There was evidence that those who were unemployed were more likely to link to care (adjusted OR (aOR) 2.09, 95% CI 1.39 to 3.14). There was an association between distance from clinic and linkage to hypertension care such that odds of presentation increased as distance to the clinic increased (aOR for linear trend in linkage with each 1 km increase in distance=1.12, 95% CI 1.05 to 1.20, p<0.001). The results of the fractional polynomial models suggested that the linear model adequately described the relationship between presentation to care and distance. After adjusting for sociodemographic and location factors, we also found strong evidence that individuals with the equivalent of stage II hypertension (aOR=2.20, 95% CI 1.63 to 2.97), and those meeting criteria for hypertensive urgency (aOR=3.07, 95% CI 2.01 to 4.67) had higher odds

of linking to care than those with the equivalent of stage I hypertension. We found similar correlates of presentation to hypertension care (age, sex, distance from clinic and employment) in sensitivity analyses with weighted models, and in models with a covariate for the number of follow-up surveys completed during 2011–2015 (online supplementary tables 3–5).

### DISCUSSION

We found very low rates of presentation to clinical care after home-based identification of elevated blood pressure in rural KwaZulu-Natal. Overall, less than one in three individuals newly identified with elevated blood pressure reported being diagnosed with hypertension or receiving treatment for elevated blood pressure within 2 years, and less than one in five had evidence of visiting a clinic for hypertension care during a 12 month period 7 years after the screen. Linkage rates were particularly low for men and young people. Notably, those employed and those closest to clinics also had poorer rates of linkage. These results highlight the important need to consider the determinants of healthcare access for NCDs in rural South Africa, and multifaceted approaches to improve linkage to care after community-based NCD screening programmes.

Studies investigating the frequency of linkage to clinical care after community- and home-based screening for chronic disease in sub-Saharan Africa have demonstrated mixed results. Most evidence has come from the HIV field, in which linkage after home-based testing in pilot studies has been highly successful,[17] although lower rates are reported in community settings.[9 18] Studies reporting clinic attendance after hypertension screening have generally shown low rates of linkage to care. For example, in a large (n=6000) health fair-based screening programme in Uganda, 41% of participants with a new positive screen

**Table 2** Factors* associated with linkage to hypertension care within 2 years after home-based diagnosis of hypertension† in 2010, among individuals who were previously undiagnosed and participated in 2011 or 2012 (n=1199) (unweighted analysis)

| | Linked to care/N (%) | Crude OR (95% CI) | Age and sex-adjusted OR (95% CI) | Adjusted OR (95% CI)‡ |
|---|---|---|---|---|
| Sociodemographic factors | | | | |
| Age group | | P<0.001 | P<0.001 | P<0.001 |
| <30 | 10/193 (5.2 %) | 1 | 1 | 1 |
| 35–44 | 40/242 (16.5%) | 3.62 (1.76 to 7.45) | 3.08 (1.49 to 6.36) | 3.32 (1.60 to 6.89) |
| 45–59 | 126/360 (35.0%) | 9.85 (5.03 to 19.30) | 8.39 (4.26 to 16.51) | 9.01 (4.57 to 17.79) |
| 60+ | 167/404 (41.3%) | 12.89 (6.62 to 25.11) | 11.61 (5.94 to 22.69) | 11.49 (5.87 to 22.46) |
| Sex | | P<0.001 | P<0.001 | P<0.001 |
| Male | 47/308 (15.3%) | 1 | 1 | 1 |
| Female | 296/891 (33.2%) | 2.76 (1.97 to 3.88) | 2.50 (1.75 to 3.57) | 2.41 (1.68 to 3.45) |
| Marital status | | P<0.001 | P=0.15 | P=0.14 |
| Single (never married) | 55/250 (22.0%) | 1 | 1 | 1 |
| Married/informal union | 178/666 (26.7%) | 1.29 (0.92 to 1.83) | 1.33 (0.91 to 1.95) | 1.35 (0.92 to 1.98) |
| Widow/separated/divorced | 110/283 (38.9%) | 2.25 (1.54 to 3.31) | 0.98 (0.64 to 1.50) | 0.99 (0.65 to 1.51) |
| Education | | P<0.001 | P=0.83 | P=0.77 |
| None | 132/363 (36.4%) | 1 | 1 | 1 |
| Less than complete secondary | 160/581 (27.5%) | 0.67 (0.50 to 0.88) | 1.10 (0.81 to 1.49) | 1.09 (0.80 to 1.49) |
| Complete secondary/above | 51/255 (20.0%) | 0.44 (0.30 to 0.64) | 1.09 (0.71 to 1.67) | 1.15 (0.75 to 1.78) |
| Employed | | P<0.001 | P=0.02 | P=0.02 |
| Yes | 31/178 (17.4%) | 1 | 1 | 1 |
| No | 312/1021 (30.6%) | 2.09 (1.39 to 3.14) | 1.71 (1.10 to 2.65) | 1.71 (1.10 to 2.65) |
| SES tertile | | P=0.307 | P=0.31 | P=0.21 |
| Low | 125/459 (27.2%) | 1 | 1 | 1 |
| Middle | 99/364 (27.2%) | 1.00 (0.73 to 1.36) | 1.09 (0.79 to 1.52) | 1.12 (0.81 to 1.56) |
| High | 115/364 (31.6%) | 1.23 (0.91 to 1.67) | 1.28 (0.93 to 1.77) | 1.34 (0.97 to 1.85) |
| Location factors | | | | |
| Residence | | P=0.04 | P=0.35 | P=0.55 |
| Urban | 10/36 (27.8%) | 1 | 1 | 1 |
| Periurban | 95/398 (23.9%) | 0.82 (0.38 to 1.75) | 0.75 (0.33 to 1.69) | 0.67 (0.29 to 1.53) |
| Rural | 238/765 (31.1%) | 1.17 (0.56 to 2.47) | 0.92 (0.41 to 2.05) | 0.63 (0.27 to 1.45) |
| Nearest clinic§ | | P<0.001 | P=0.001 | P=0.001 |
| Per each km of distance | | 1.15 (1.08 to 1.23)¶ | 1.13 (1.05 to 1.21)¶ | 1.12 (1.05 to 1.20)¶ |
| 0-<1.5 | 56/263 (21.3%) | | | |
| 1.5–2.5 | 71/269 (26.4%) | | | |
| >2.5–3.9 | 93/310 (30.0%) | | | |
| >3.9 | 123/357 (34.5%) | | | |
| Clinical factors | | | | |
| BMI category | | P<0.001 | P=0.07 | P=0.13 |
| <25 kg/m$^2$ | 70/344 (20.3%) | 1 | 1 | 1 |
| 25 –<30 kg/m$^2$ | 62/229 (27.1%) | 1.45 (0.98 to 2.15) | 1.08 (0.71 to 1.64) | 1.17 (0.76 to 1.81) |
| ≥30 kg/m$^2$ | 110/301 (36.5%) | 2.25 (1.59 to 3.21) | 1.52 (1.03 to 2.24) | 1.51 (1.00 to 2.26) |
| Hypertension stage** | | P<0.001 | P<0.001 | P<0.001 |
| Stage I | 142/730 (19.5%) | 1 | 1 | 1 |
| Stage II | 134/342 (39.2%) | 2.67 (2.01 to 3.54) | 2.22 (1.65 to 2.99) | 2.20 (1.63 to 2.97) |
| Hypertension urgency | 67/127 (52.8%) | 4.62 (3.12 to 6.85) | 3.12 (2.06 to 4.74) | 3.07 (2.01 to 4.67) |

**Table 2** Continued

| | Linked to care/N (%) | Crude OR (95% CI) | Age and sex-adjusted OR (95% CI) | Adjusted OR (95% CI)‡ |
|---|---|---|---|---|
| Self-report of diabetes†† | | P=0.19 | P=0.44 | P=0.47 |
| No | 339/1191 (28.5%) | 1 | 1 | 1 |
| Yes | 4/8 (50.0%) | 2.51 (0.62 to 10.11) | 1.78 (0.41 to 7.70) | 1.75 (0.38 to 8.15) |
| Self-report of TB†† | | P=0.66 | P=0.96 | P=0.72 |
| No | 332/1156 (28.7%) | 1 | 1 | 1 |
| Yes | 11/43 (25.6%) | 0.85 (0.43 to 1.71) | 1.02 (0.49 to 2.13) | 1.15 (0.54 to 2.46) |

*All characteristics are based on 2010 survey.
†Hypertension defined as systolic BP ≥140 mm Hg or diastolic BP ≥90 mm Hg, in an average of two readings.
‡Sociodemographic factors adjusted for age group, sex and employment. Location factors adjusted for age group, sex, employment and distance from nearest clinic as continuous covariate. Clinical factors adjusted for age group, sex, employment, distance from nearest clinic and hypertension stage.
§Quartiles based on distribution in all individuals who were eligible for 2010 survey. Fit as continuous covariate; n (%) linked in each distance quartile shown for information only.
¶OR for linear trend in linkage with each 1 km increase in distance.
**Stage I: systolic BP 140–159 or diastolic BP 90–99; stage II: systolic BP 160–179 or diastolic BP 100–119; hypertension urgency: systolic BP ≥180 or diastolic BP ≥120.
††Reports being diagnosed in the past 12 months or currently on treatment.
BMI, body mass index; BP, blood pressure; TB, tuberculosis.

for elevated blood pressure linked to care.[9] A pilot study in Kenya that compared a home-based (n=236) with a community-based health fair booth approach (n=346) for hypertension and diabetes screening, found equally low rates of linkage to care (30%) with both strategies.[19] A smaller study in Kenya yielded higher linkage rates (74%, n=120) after community group-initiated blood pressure screening.[20] Interpreting these contrasting results must be done with attention to the selection criteria of each. Whereas our procedures were home-based, the larger study from Uganda included self-referring individuals who had attended a health fair, and the Kenyan study operated through a peer microfinance programme, in which NCD screening services were paired with agribusiness advice within preorganised community groups. In the prior report most similar to ours, a large programme in Malawi (n=27 305) that provided clinical referrals after home-based testing reported a 59% linkage rate within 2 weeks of a diagnosis of hypertension, although 30% of participants were already on treatment at the time of referral.[21] Moreover, approximately 50% dropped out of care within 6 months of linkage.

Predictors of presentation to care in our study reinforce much of the literature on healthcare access and engagement among vulnerable populations in sub-Saharan Africa. Lower engagement by younger individuals and men is a well-established phenomenon and a public health challenge for the region.[22 23] Although it did not reach statistical significance, we also found evidence that those with greater social support, as evidenced by having a cohabitating partner, had a 35% increased odds of presenting to care. An unexpected finding was that those who were unemployed and those further from clinic were more likely to link to care. This finding contrasts with much of the data from the region on how distance from health services impacts linkage to and retention

in care.[24–26] We hypothesise that these results illustrate competing demands between obligations to work and to access healthcare. Notably, a similar phenomenon was found in the Malawi home-based NCD screening study, in which rural participants had more than twice the odds of linkage to NCD care than their urban counterparts, and the most common reason stated for failure to link to care in urban areas was being too busy to attend clinic; reported in 34% of those not linked to care.[21] Employment is higher in male than in female South African blacks,[27] and so may also contribute to the gender difference in linkage.

Our data do offer multiple potential strategies to improve linkage to care after home-based NCD screening. For example, a notable distinction between many community-based HIV and NCD diagnostic programmes is the degree of counselling and referral services provided after diagnosis. Hypertension referral services in our programme, and many others in the region are often limited to distribution of results and referral forms. The potential beneficial effects of more comprehensive health and lifestyle counselling on the success of linkage after an elevated blood pressure screen should be actively explored in future work. For example, decades of standardising in-depth HIV counselling services, and additional facilitated linkage strategies have significantly improved rates of linkage after a new HIV diagnosis.[28] Pilot studies of enhanced referral after community based NCD diagnoses have also shown promise in vulnerable populations in the USA,[29 30] and warrant investigation on a larger scale elsewhere. Similarly, our finding and that of others that hypertension linkage was less common in those employed presents a potential opportunity to consider expanded clinic service hours and/or community based management to improve NCD care in the region. Endeavours, such

as the Centralised Chronic Medication Dispensing and Distribution recently launched by the KwaZulu-Natal Department of Health, seek to overcome such barriers by delivering medicines to peoples' homes and workplaces, or setting up community-based medicine pick-up points. Evaluations of the efficacy and sustainability of such programmes will be of high importance to the field. Finally, our results, and particularly the high prevalence of elevated blood pressure and low rates of care sinking after notification of such, highlight the critical importance of risk factor modification as part of public health strategies. Such programmes, including smoking cessation, healthy diet and exercise promotion, which are recommended by South African Department of Health hypertension control policies, have potential to significantly impact health and must also remain a cornerstone of population hypertension control programmes.[31 32]

Our study is strengthened by a large sample size and the use of a home-based testing paradigm. The primary limitation to our analysis is the relatively low response rate in the initial hypertension screen and participation in subsequent surveys. We accounted for this limitation by comparing characteristics between participants and non-participants, and by using IPW techniques to make population level inferences. We hypothesise that the low participation in this instance was due to the placement of the screening activity within a routine annual survey, as opposed to a stand-alone health promotion. Nonetheless, it should be noted that a similar response rate for health interventions would also pose a challenge to their use for population-wide screening programmes. Our study is also limited by a use of self-report to detect linkage to clinical care over the first 2 years of observation and clinic records 7 years later which may lead to misestimation of outcomes. For example, participants who presented to clinic and had a normal blood pressure might not be detected by self-report of a new diagnosis of hypertension or with use of clinical records to 7 years after the initial screen. Our study also did not investigate if and how supply-side considerations, such as staffing, wait times and drug availability might have contributed to the low rates of linkage. Although hypertension care and treatment is provided free of charge in South Africa in health clinics in the public sector, these factors have been demonstrated to affect engagement in care elsewhere in the country.[33 34] Finally, our results should be considered in the context of the low-resource, rural sub-Saharan Africa study setting, but are unlikely to generalise more broadly to urban or higher resource regions.

In summary, we found very low rates of linkage to care after a population-level, home-based hypertension screen in rural KwaZulu-Natal. Strategies focused on increased demand generation, particularly for younger individuals and men, augmented referral and linkage programmes, and efforts to enhance the convenience of service delivery, particularly to employed people, should be evaluated to improve NCD care access after community based testing in the region.

**Author affiliations**
¹Medicine, Massachusetts General Hospital, Boston, Massachusetts, USA
²Department of Medicine, Harvard Medical School, Boston, Massachusetts, USA
³Africa Health Research Institute, KwaZulu-Natal, South Africa
⁴Epidemiology and Medical Statistics, London School of Tropical Medicine and Hygiene, London, UK
⁵Bordeaux Population Health Research Center, University of Bordeaux, Bordeaux, France
⁶University College London, London, UK
⁷Institute of Global Health, University of Heidelberg, Heidelberg, Germany
⁸Harvard School of Public Health, Boston, Massachusetts, USA
⁹Population Health and Primary Care, University of East Anglia, Norwich, UK

**Contributors** MJS: conceived of the project, wrote the first draft and contributed to the data analysis. KB: led the data analysis, contributed to manuscript production. JO-G: helped conceive of the project and contributed to manuscript production. DP: contributed to data collection, project conception and contributed to manuscript production. OK: contributed to data collection, project conception and contributed to manuscript production. EBW: contributed to data collection and project conception. PM: contributed to data collection, project conception and contributed to manuscript production. FT: contributed to data collection and project conception. KH: contributed to data collection and project conception. TB: contributed to data collection, project conception and contributed to manuscript production. MB: contributed to project conception, data analysis and contributed to manuscript production. All authors reviewed the manuscript for editorial contributions and approved of the final version.

**Funding** The Africa Health Research Institute receives funding from the UK Wellcome Trust grant 082384/Z/07/Z. MJS receives funding from the National Institutes of Health (K23 MH099916, P30 30AI060354). EBW receives funding from the National Institutes of Health (K08 AI118538). The population cohort is partly funded by the South African Department of Science and Technology through the SA-MRC hosted South African Population Research Infrastructure Network.

**Competing interests** None declared.

**Patient consent** Obtained.

**Ethics approval** Biomedical Research Ethics Committee of the University of KwaZulu-Natal, South Africa.

**Provenance and peer review** Not commissioned; externally peer reviewed.

**Data sharing statement** Data for this analysis is available by emailing Mark Siedner (msiedner@mgh.harvard.edu) and Kobus Herbst (kherbst@ahri.org).

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
