## [Reviewer comments · BMJ Open]

ARTICLE DETAILS

TITLE (PROVISIONAL)	Linkage to primary care after home-based blood pressure screening in rural Kwazulu-Natal, South Africa: A population-based cohort study
AUTHORS	Siedner, Mark; Baisley, Kathy; Orne-Gliemann, Joanna; Pillay, Deenan; Koole, Olivier; Wong, Emily; Matthews, Philippa; Tanser, Frank; Herbst, Kobus; Barnighausen, Till; Bachmann, Max

VERSION 1 – REVIEW

REVIEWER	Mathieu Maheu-Giroux McGill University, Canada
REVIEW RETURNED	07-May-2018

GENERAL COMMENTS	The article by Mark Siedner and colleagues investigates linkage to care following a community-based screening programs for hypertension in rural KwaZulu-Natal, South Africa. The paper is well-written, the methods and analyses are generally clearly described, and the description of the study's strengths/limitations appropriate. The paper is important and most of my comments are relatively minor and/or can be easily addressed. Importantly, providing more details on the counseling/referral services offered as part of the home-based screening program is warranted. Important comments: 1. The authors adjusted for non-participation in the baseline 2010 home-based screening program using inverse probability weighting (IPW). Non-response in the follow-up surveys among those newly diagnosed was important (30% non-response or loss to follow-up). Why haven't the authors adjust for this too using IPW (only 1,199 out of 1,706 undiagnosed hypertensive included at follow-up)? I also recommend presenting the unadjusted estimates of the proportion that linked to care in the paper (i.e., 343/1,199=28.6% linked to care within two years).2. Page 2, line 42. The authors assert that the population encompassed by the Demographic Health Surveillance area in KwaZulu-Natal is "under-studied". Given that this population has been subjected to bi-annual surveillance for more than a decade, I would not qualify it as being "under-studied" – quite to the contrary.3. "Blood pressure screening and referral methods" (page 5). Could the authors provide more information on the referral process following a positive hypertension screen? Was there any counselling on the importance of seeking appropriate care? Was this home-based screening programs conceptualized as such or part of the on-going data collection of the DHS and, for ethical
---

	reasons, referral forms had to be handed out to patients with high blood pressure? If that is the case, that could explain the very low linkage rates – that is was not a home-based screening programs per se but, rather, more akin to a simple blood pressure survey with referral, if warranted. The authors briefly touch upon this in the discussion, but a thorough description (with specific details on length/quality of counseling) of this home-based screening program in the methods is warranted. 4. I am not sure I understand how the outcome was defined (on page 6). The home-based screening program was performed in 2010. There was a follow-up survey in 2011 and the linkage to care outcome was based on self-reports of hypertension in the last 12 months. Could it be that some individuals reported that their initial diagnosis was from the home-based screening and that these people, according to the definitions of linkage used by the authors, are counted as newly linked? 5. The authors examined if self-reports of diabetes or TB are associated with linkage to care. Why not include HIV, which is prevalent in the area? The authors could have used self-reported knowledge of HIV status. Alternatively, Hlabisa has information on all HIV positive individuals linked to HIV treatment and care in the area. 6. The discussion focuses a lot on approaches to increase linkage to care. More generally, I found that the low participation rates in the home-based screening program (45%) much more concerning from a programmatic point of view. Low participation would compromise the effectiveness (and costs) of such screening programs as much as low linkage but the reasons for this low participation are not discussed. For example, are higher rates of participation expected in other (less intensively surveyed/studies) areas of South Africa?. More generally, alternative population approaches to address the underlying risk factors for high blood pressure in this population should probably be mentioned in the discussion (à la Geoffrey Rose). Minor comments:  -Add the name of the country in the title. -Proportions are sometime presented in the manuscript with one decimal point, sometimes they are rounded to the nearest integer. Please standardize. -Line 21. Typo? “gained traction recently as a mean(s)”. Remove “s” from means. -Line 78. Could the authors add (in parentheses) the strata used to categorize the variables included in the logistic regression model used to calculate IPW. -Line 85. How was socio-economic status measured? An asset-based index? Please describe in more details. -Line 102. Typo? “verified in a(n) SQL database”. Remove “n” from “an”. -Line 173. Typo? “95%C6.62”. Add “I” in “95%CI”.
--	--

	-Line 184. Typo? "aOR=3.07, CI=2.01". Add "95%" before "CI". -Table 1. The sample sizes for each column at the top sometimes separates 1,000 with a comma, sometimes not. Please standardized. -Table 1. 3,074/11,694 = 26.3% (and not 26.2%). -Table 2. Presenting linkage to care by distance to the nearest clinics by categories and then using a linear term for the regression is misleading. One could interpret that the OR is for the 2.5-2.9 km category. -There is a lot of emphasis throughout the manuscript on "statistical significance". However, there is nothing special about a threshold p-value of 0.05 and a more nuanced interpretation of effect size estimates would be welcomed (especially given the limitations of the study).
--	---

REVIEWER	Paul Rheeder University of Pretoria, South Africa
REVIEW RETURNED	22-May-2018

GENERAL COMMENTS	Excellent paper on a very important topic. Home based visits will be an important feature in the South African health care system as it is in other developing countries and the challenges and opportunities this presents needs studies such as this one. Excellent work.
---

REVIEWER	Sonak Pastakia AMPATH/Moi University/Purdue University (Kenya)
REVIEW RETURNED	06-Jul-2018

GENERAL COMMENTS	Summary-This article describes in great detail short and long term linkage patterns for patients receiving hypertension screening in rural KZN. This is an understudied but crucial aspect of providing care to patients with hypertension in rural under resourced settings like KZN. While linkage to care is often overlooked or looked to over a very short period of time, this article highlights the linkage patterns over a much longer period of time and stimulates much needed research in this aspect of care. Intro -I would like to see more details on the clinic capacity for managing NCDs specifically. (i.e. is care free, are meds for htn usually available, are wait times reasonable and staff generally available, etc.). These aspects could certainly have a significant impact on linkage. A brief statement on these aspects of care delivery would help the reader contextualize this program. Abstract. It mentions SBP>140 and DBP>90 instead of SBP greater than or equal to 140 or DBP greater than or equal to 90. Please make sure all blood pressure ranges mentioned in the article are consistent and appropriate with regards to being listed as ">" or greater than or equal to. Methods pg 6, line 51- Can you provide justification for your strategy of checking 3 bps and only using the last two? Is that approach
---

	supported by a national guideline or study that you could reference? -are there private sector options for care? are you able to differentiate where they get care from based on the questionnaire responses? -The thoroughness of their methods for assessing linkage are noteworthy and should be commended. Verifying results in the manner they have done sets a very nice precedent for other studies investigating these issues. -pg 8, line 86, why didn't you also check for HIV status? Seems likely that would be a predictive variable especially since those populations are at greater risk of htn and KZN tends to have a higher prevalence of HIV. Results -pg 11, line 174, can you highlight which factors you adjusted for? Discussion -When you discuss the study in reference 18, I would recommend delving a little deeper and explain that screening was done in a community based fashion and care delivery was also linked to the microfinance and agriculture based community groups. Instead of telling patients that they had to go to clinic, they had the option of receiving care from clinicians who would visit their groups and provide care directly there. Space permitting, it would be nice to mention that a similar region in this area previously reported a 30% linkage rate similar to this study and this increase was subsequently observed after contextualizing care which incorporated ag and microfinance support. (see Pastakia et al. Globalization and Health, Screening for diabetes and hypertension in a rural low income setting in western Kenya utilizing home-based and community-based strategies) Limitations How confident are you of the accuracy of the public health records and your ability to reliably link results. Seems like a massive undertaking and would rely on very accurate records. Does the clinic follow some of the same standards as the DHS in terms of collecting research quality data. For my own edification, how was the research team able to track 1706 patients after 2 years... 1199 participated in the survey after 2 years of being diagnosed, but then the number of participants dropped significantly to 222 after 3-5 years (page 24)... In conclusion, I thoroughly enjoyed reviewing this paper and look forward to seeing the responses to the small revisions requested
--	---

VERSION 1 – AUTHOR RESPONSE

Reviewer: 1

The article by Mark Siedner and colleagues investigates linkage to care following a community-based screening programs for hypertension in rural KwaZulu-Natal, South Africa. The paper is well-written, the methods and analyses are generally clearly described, and the description of the study's strengths/limitations appropriate. The paper is important and most of my comments are relatively minor and/or can be easily addressed. Importantly, providing more details on the counseling/referral services offered as part of the home-based screening program is warranted.

Important comments:

1. The authors adjusted for non-participation in the baseline 2010 home-based screening program using inverse probability weighting (IPW). Non-response in the follow-up surveys among those newly diagnosed was important (30% non-response or loss to follow-up). Why haven't the authors adjust for this too using IPW (only 1,199 out of 1,706 undiagnosed hypertensive included at follow-up)? I also recommend presenting the unadjusted estimates of the proportion that linked to care in the paper (i.e., $343/1,199=28.6\%$ linked to care within two years).

Response

We agree with the concern about loss to follow-up and potential for bias due to missing data. To account for this, we: 1) compared characteristics between those who did and did not undergo blood pressure screening in 2010; 2) ran a sensitivity analysis in which we used IPWs of screening in the models (similar to the request by this reviewer) and 3) ran a sensitivity analysis including the number of screening visits participated in during 2011-2015. We found similar correlates of linkage to hypertension care in these models, which are presented in Supplemental Tables 3-5. The methods for these procedures are reported in the methods section (lines 203-208):

...we compared characteristics of eligible individuals who did and did not complete blood pressure screening in 2010. Next, we compared characteristics of those who participated in a subsequent health survey and those who did not, either because of refusal, out-migration, or death. Finally, we conducted sensitivity analyses in which we: 1) used IPWs of screening in the models; and 2) added a covariate to indicate the number of individual health surveys participated in during 2011-2015.

In response to the reviewer's request here, we have added the unadjusted prevalence of linkage to hypertension care to the manuscript as well (lines 272-273):

The crude and IPW-adjusted population prevalence of linkage to hypertension care within two years of the blood pressure screen was 28.6% (95%CI=26.1-31.2%), and 26.9% (95%CI=24.5-29.4%), respectively

2. Page 2, line 42. The authors assert that the population encompassed by the Demographic Health Surveillance area in KwaZulu-Natal is "under-studied". Given that this population has been subjected to bi-annual surveillance for more than a decade, I would not qualify it as being "under-studied" – quite to the contrary.

Response

Thank you for this suggestion. We did not mean to suggest that the population itself was under-studied, but that hypertension as a condition and its management have received relatively little attention. We have reworded that section as follows (lines 83-84):

- Assesses a population in rural sub-Saharan Africa who are noted to have high prevalence of hypertension but with little corresponding data about linkage to care after diagnosis

3. "Blood pressure screening and referral methods" (page 5). Could the authors provide more information on the referral process following a positive hypertension screen? Was there any counselling on the importance of seeking appropriate care? Was this home-based screening programs conceptualized as such or part of the on-going data collection of the DHS and, for ethical reasons, referral forms had to be handed out to patients with high blood pressure? If that is the case, that could explain the very low linkage rates – that is was not a home-based screening programs per

se but, rather, more akin to a simple blood pressure survey with referral, if warranted. The authors briefly touch upon this in the discussion, but a thorough description (with specific details on length/quality of counseling) of this home-based screening program in the methods is warranted.

Response

We agree with the reviewer that this activity was done primarily for research purposes and that might have affected the downstream success rate. We note however that the repeated measurement of blood pressure in line with WHO Steps guidelines, provision of a paper copy of results to individuals, and encouragement of referral for those with abnormal results would meet the definition of a screening event in many settings. To account for the valid concern raised by the reviewer, we have added detail to the description of the screening and referral process (lines 143-146):

All individuals were given a copy of their blood pressure results, along with body mass index, on a health results card. Those with elevated blood pressure were counselled by study staff to seek care at their preferred local public health clinic with a copy their screening results.

We have also added more context this consideration to the discussion (lines 367-372):

Hypertension referral services in our program and many others in the region are often limited to distribution of results and referral forms. The potential beneficial effects of more comprehensive health and lifestyle counseling on the success of linkage after an elevated blood pressure screen should be actively explored in future work. For example, decades of standardizing in-depth HIV counseling services, and additional facilitated linkage strategies have significantly improved rates of linkage after a new HIV diagnosis.

4. I am not sure I understand how the outcome was defined (on page 6). The home-based screening program was performed in 2010. There was a follow-up survey in 2011 and the linkage to care outcome was based on self-reports of hypertension in the last 12 months. Could it be that some individuals reported that their initial diagnosis was from the home-based screening and that these people, according to the definitions of linkage used by the authors, are counted as newly linked?

Response

The home based survey was conducted in 2010. All participants were part of the greater demographic health and surveillance (DHS) site, and are seen annually for home-based surveys. We defined the primary outcome of interest as linkage to care, as defined by report of attending a health facility for hypertension in the last 12 months at either of the 2011 or 2012 surveys in the subsequent two years. We have clarified this with additional text in the methods section (149 – 157):

For our primary outcome of interest, we defined successful linkage to care for hypertension as self-reported linkage within two years of a positive home-based hypertension screen. To assess this outcome, we used data from the two subsequent, annual home-based health surveys in 2011 and 2012. All members of the initial blood pressure screening survey were members of the demographic health and surveillance site study population, and are seen annually at home for data collection. In each annual health survey, respondents were asked if: 1) they have been diagnosed with hypertension in the past 12 months; 2) if they have ever received hypertension treatment; and 3) if they are currently being treated for hypertension. We defined successful linkage to care by a positive response to any of these three questions in either 2011 or 2012.

We also investigated whether our outcome of interest might be over-estimating hypertension linkage due to misclassification of the initial blood pressure screen as the “new” diagnosis of hypertension at the next survey visit. However, of the 343 new diagnoses in the 2011 and 2012 surveys, 218 (64%) took place in 2011. But 191 of them reported currently being on treatment, leaving only 27 out of 343

total linkages (8%), they have any potential of being misclassified as new diagnoses. Therefore, we feel there is a low risk of over-classification. We have added these results to the text to provide more clarity (lines 273-276).

Of the 343 total new linkages reported, 218 (64%) and 135 (36%) were reported in 2011 and 2012, respectively. Of the 218 new linkages reported in 2011, the majority (191, 88%) reported initiating anti-hypertensive therapy

5. The authors examined if self-reports of diabetes or TB are associated with linkage to care. Why not include HIV, which is prevalent in the area? The authors could have used self-reported knowledge of HIV status. Alternatively, Hlabisa has information on all HIV positive individuals linked to HIV treatment and care in the area.

Response

We fully agree that HIV diagnosis would have added important information to this analysis. Unfortunately, at the time of the hypertension survey, we did not inquire to individuals in the DHS survey about their HIV serostatus, only their testing history. The survey also predated our current data linkage that connects DHS participants to clinic and hospital records. We have added a sentence to explain this limitation (lines 190-191):

Self-reported data on HIV diagnosis was not collected at the time of the hypertension assessment, so could not be included in the analysis.

6. The discussion focuses a lot on approaches to increase linkage to care. More generally, I found that the low participation rates in the home-based screening program (45%) much more concerning from a programmatic point of view. Low participation would compromise the effectiveness (and costs) of such screening programs as much as low linkage but the reasons for this low participation are not discussed. For example, are higher rates of participation expected in other (less intensively surveyed/studies) areas of South Africa?. More generally, alternative population approaches to address the underlying risk factors for high blood pressure in this population should probably be mentioned in the discussion (à la Geoffrey Rose).

Response

Thank you for this comment. We agree that low participation in community and home based surveys could be a limitation of this approach, and have added it to the discussion section. Notably, the screening program took place in the midst of a routine DHS survey, which had been active for the past 10 years, so we hypothesize in this scenario it was possibly due to a lack of novelty than lack of interest (lines 392 – 399):

Our study is strengthened by a large sample size and the use of a home-based testing paradigm. The primary limitation to our analysis is the relatively low response rate in the initial hypertension screen and participation in subsequent surveys. We accounted for this limitation by comparing characteristics between participants and non-participants, and by using inverse probability weighting techniques to make population level inferences. We hypothesize that the low participation in this instance was due to the placement of the screening activity within a routine annual survey, as opposed to a stand-alone health promotion. Nonetheless, it should be noted that a similar response rate for health interventions would also pose a challenge to their use for population-wide screening programs.

We also agree that we gave too little attention to the importance of prevention of hypertension through risk factor reduction, as opposed to the downstream screen and treat method that we have studied and discuss at length. We have added content to the discussion to address this shortcoming (lines 385-390):

Finally, our results, and particularly the high prevalence of elevated blood pressure and low rates of care seeking after notification of such, highlight the critical importance of risk factor modification as part of public health strategies. Such programs, including smoking cessation, healthy diet, and exercise promotion, which are recommended by South African Department of Health hypertension control policies, have potential to significantly impact health, and must also remain a cornerstone of population hypertension control programs.(1, 2)

Minor comments:

-Add the name of the country in the title.

Response:

We have done as suggested.

-Proportions are sometime presented in the manuscript with one decimal point, sometimes they are rounded to the nearest integer. Please standardize.

Response: Thank you for noting this. We have corrected this to make all with one decimal point.

-Line 21. Typo? "gained traction recently as a mean(s)". Remove "s" from means.

Response:

We believe this is correctly written as is, but happy to reconsider if editor/reviewer feels otherwise.

-Line 78. Could the authors add (in parentheses) the strata used to categorize the variables included in the logistic regression model used to calculate IPW.

Response:

These are listed in lines 176-179:

To calculate the weights, we fit a logistic regression model with completion of blood pressure screening in 2010 as the outcome of interest and included age strata, sex, education level and place of residence (urban, peri-urban, or rural) as predictors, based on information routinely collected in the household-level survey.

-Line 85. How was socio-economic status measured? An asset-based index? Please describe in more details.

Response:

We conducted a principal components analysis of households asset ownership following the method recommend by Filmer and Pritchett asset index. We have added this description to the text (lines 183-186):

Potential determinants of linkage were examined at three levels: sociodemographic factors (age, sex, educational attainment, employment status and socioeconomic status, which was estimated using principal components analysis of household asset ownership following the method of Filmer and Pritchett(3))

-Line 102. Typo? "verified in a(n) SQL database". Remove "n" from "an".

Response:

We believe this was correct as written. See for example: <https://www.aje.com/en/arc/editing-tip-indefinite-article-use-abbreviations/>

-Line 173. Typo? "95%CI". Add "I" in "95%CI".

Response:

Thank you for catching this. We have made that change.

-Line 184. Typo? "aOR=3.07, CI=2.01". Add "95%" before "CI".

Response:

Thank you for catching this. We have made that change.

-Table 1. The sample sizes for each column at the top sometimes separates 1,000 with a comma, sometimes not. Please standardized.

Response:

Thank you for catching this. We have made that change.

-Table 1. $3,074/11,694 = 26.3\%$ (and not 26.2%).

Response:

Thank you for catching this. We have made that change.

-Table 2. Presenting linkage to care by distance to the nearest clinics by categories and then using a linear term for the regression is misleading. One could interpret that the OR is for the 2.5-2.9 km category.

Response:

We have changed the format of Table 2 to have a separate line to demonstrate that the regression term is for each km of distance

-There is a lot of emphasis throughout the manuscript on "statistical significance". However, there is nothing special about a threshold p-value of 0.05 and a more nuanced interpretation of effect size estimates would be welcomed (especially given the limitations of the study).

Response:

We agree completely with the reviewer on this point. It is for this reason that we produced graphics to demonstrate the absolute level of linkage to care by age and sex. The term "statistical significance" is only mentioned once in the discussion. In that instance, we have add an effect size to better classify the magnitude of the effect (lines 351-352):

Although it did not reach statistical significance, we also found evidence that those with greater social support, as evidenced by having a cohabitating partner, had a 35% increased odds of presenting to care

Reviewer: 2

Excellent paper on a very important topic. Home based visits will be an important feature in the South African health care system as it is in other developing countries and the challenges and opportunities this presents needs studies such as this one. Excellent work.

Response:

We thank the reviewer for their positive feedback.

Reviewer: 3

Summary-This article describes in great detail short and long term linkage patterns for patients receiving hypertension screening in rural KZN. This is an understudied but crucial aspect of providing care to patients with hypertension in rural under resourced settings like KZN. While linkage to care is often overlooked or looked to over a very short period of time, this article highlights the linkage patterns over a much longer period of time and stimulates much needed research in this aspect of care.

Intro

-I would like to see more details on the clinic capacity for managing NCDs specifically. (i.e. is care free, are meds for htn usually available, are wait times reasonable and staff generally available, etc.). These aspects could certainly have a significant impact on linkage. A brief statement on these aspects of care delivery would help the reader contextualize this program.

Response:

We agree that the lack of supply-side factors in our analysis is an important limitation that was not addressed in the initial manuscript. We feel speculating about the availability of drugs and wait times without collecting primary data ourselves (or knowing of any in our district) might be presumptive. As such, we have added text to the limitations section to describe that hypertension care is provided free of care, but noting the gap in assessment of supply side factors in our analysis (lines 403-407):

Our study also did not investigate if and how supply-side considerations, such as staffing, wait times, and drug availability might have contributed to the low rates of linkage. Although hypertension care and treatment is provided free of charge in South Africa in health clinics in the public sector, these factors have been demonstrated to affect engagement in care elsewhere in the country.(4, 5)

Abstract.

It mentions SBP>140 and DBP>90 instead of SBP greater than or equal to 140 or DBP greater than or equal to 90. Please make sure all blood pressure ranges mentioned in the article are consistent and appropriate with regards to being listed as ">" or greater than or equal to.

Response:

Thank you. We have specified that it is indeed greater to or equal to 140 and 90.

Methods

pg 6, line 51- Can you provide justification for your strategy of checking 3 bps and only using the last two? Is that approach supported by a national guideline or study that you could reference?

Response:

This is recommended by the WHO Steps protocol. See here, page 3-5-5:
http://www.who.int/ncds/surveillance/steps/STEPS_Manual.pdf

-are there private sector options for care? are you able to differentiate where they get care from based on the questionnaire responses?

Response:

We measured our primary outcome of interest by asking participants to report if they had been

diagnosed with or treated for hypertension, but we do not specify the location. In a secondary outcome, we assessed whether they were in clinical care using clinical linkage data to the local public clinics. This outcome does not include linkage to care to the private sector. We have added this information to the methods sections (lines 198-203):

confirmation of a clinic appointment for hypertension in 2017 at any of the 11 local public sector clinics, among those who remained a resident in the catchment area

-The thoroughness of their methods for assessing linkage are noteworthy and should be commended. Verifying results in the manner they have done sets a very nice precedent for other studies investigating these issues.

Response:
Thank you.

-pg 8, line 86, why didn't you also check for HIV status? Seems likely that would be a predictive variable especially since those populations are at greater risk of htn and KZN tends to have a higher prevalence of HIV.

Response:
We agree with the reviewer that this would have been an important piece of data to include. Unfortunately, at the time of the initial blood pressure survey, HIV testing was done anonymously, and we did not ask about self-reported HIV status. We have added this information to the methods section (lines 190-191):

Self-reported data on HIV diagnosis was not collected at the time of the hypertension assessment, so could not be included in the analysis.

Results

-pg 11, line 174, can you highlight which factors you adjusted for?

Response:
The model selection for the adjusted models are listed in Table 2 and in the methods section (lines 183-192):

Potential determinants of linkage were examined at three levels: sociodemographic factors (age, sex, educational attainment, employment status and socioeconomic status, which was estimated using principal components analysis of household asset ownership following the method of Filmer and Pritchett(3)); geographical factors (distance from clinic, urban versus rural residency); and clinical factors (body mass index, elevated blood pressure severity [defined using hypertension stages as a) stage I: systolic 140-160 and diastolic 80-100; b) stage II: systolic 160-<180 or diastolic 100-<120; or c) hypertensive urgency: systolic \geq 180 or diastolic \geq 120], self-report of diabetes, self-report of tuberculosis)(6). Self-reported data on HIV diagnosis was not collected at the time of the hypertension assessment, so could not be included in the analysis. Sociodemographic and clinical factors whose age- and sex-adjusted association with linkage was significant at $p < 0.10$ were included in a final adjusted multivariable model

To save from listing them all, but specify the categories of adjusted variables we have added the following text to the results section (line 286):

In models adjusted for socioeconomic, geographic, and clinical factors, we found strong evidence that...

Discussion

-When you discuss the study in reference 18, I would recommend delving a little deeper and explain that screening was done in a community based fashion and care delivery was also linked to the microfinance and agriculture based community groups. Instead of telling patients that they had to go to clinic, they had the option of receiving care from clinicians who would visit their groups and provide care directly there. Space permitting, it would be nice to mention that a similar region in this area previously reported a 30% linkage rate similar to this study and this increase was subsequently observed after contextualizing care which incorporated ag and microfinance support. (see Pastakia et al. Globalization and Health, Screening for diabetes and hypertension in a rural low income setting in western Kenya utilizing home-based and community-based strategies)

Response:

We have included text in the discussion to discuss the secondary activities including in the Kenya study (lines 339-342):

Whereas our procedures were home-based, the larger study from Uganda included self-referring individuals who had attended a health fair, and the Kenyan study operated through a peer microfinance program, in which NCD screening services were paired with agribusiness advice within pre-organized community groups.

Thank you for pointing out the second study. We have also now included discussion of this in the manuscript (lines 335-37)

A pilot study in Kenya that compared a home-based (n=236) with a community-based healthfair booth approach (n=346) for hypertension and diabetes screening, found equally low rates of linkage to care (30%) with both strategies(7)

Limitations

How confident are you of the accuracy of the public health records and your ability to reliably link results. Seems like a massive undertaking and would rely on very accurate records. Does the clinic follow some of the same standards as the DHS in terms of collecting research quality data. For my own edification, how was the research team able to track 1706 patients after 2 years... 1199 participated in the survey after 2 years of being diagnosed, but then the number of participants dropped significantly to 222 after 3-5 years (page 24)...

Response:

Regarding the 222 in years 3-5, we apologize for this confusion. Those are 222 people that did not participate in 2011 or 2012, but were seen in 2013-2015. It does not indicate that only 222 people participated in those years. We have edited the figure to clarify that point.

Full details about our demographic health and surveillance site data collection methods have been published previously (8). For the clinic link data, we hire staff who work at all of the clinics in the catchment area at the clinic registration. When a patient arrives, they are identified by their DHS ID, or if unknown by names and dates of birth of themselves and resident family members. Although we feel this linkage system provides a unique strength of our DHS data system by allowing evaluation of health system interactions, we also acknowledge that this system will have missed visits and misidentifications. This is partially why we considered the clinic linkage outcome as a secondary outcome in a sensitivity analysis.

In conclusion, I thoroughly enjoyed reviewing this paper and look forward to seeing the responses to the small revisions requested

Response:
Thank you.

References Cited:

1. Rose G. Sick individuals and sick populations. *International journal of epidemiology*. 2001;30(3):427-32; discussion 33-4.
2. South African Health Review. Health Systems Trust. Durban, South Africa. 2016. Available at: <https://www.health-e.org.za/wp-content/uploads/2016/05/South-African-Health-Review-2016.pdf>; Accessed 8th January 2018.
3. Filmer D, Pritchett LH. Estimating wealth effects without expenditure data--or tears: an application to educational enrollments in states of India. *Demography*. 2001;38(1):115-32.
4. Wollum A, Gabert R, McNellan CR, Daly JM, Reddy P, Bhatt P, et al. Identifying gaps in the continuum of care for cardiovascular disease and diabetes in two communities in South Africa: Baseline findings from the HealthRise project. *PLoS One*. 2018;13(3):e0192603.
5. Ameh S, Klipstein-Grobusch K, D'Ambruso L, Kahn K, Tollman SM, Gomez-Olive FX. Quality of integrated chronic disease care in rural South Africa: user and provider perspectives. *Health policy and planning*. 2017;32(2):257-66.
6. Glanz K, Rimer BK, Viswanath K. *Health behavior and health education: Theory, Research and Practice*. 4th ed. San Francisco, CA: Jossey-Bass; 2008.
7. Pastakia SD, Ali SM, Kamano JH, Akwanalo CO, Ndege SK, Buckwalter VL, et al. Screening for diabetes and hypertension in a rural low income setting in western Kenya utilizing home-based and community-based strategies. *Globalization and health*. 2013;9:21.
8. Tanser F, Hosegood V, Barnighausen T, Herbst K, Nyirenda M, Muhwava W, et al. Cohort Profile: Africa Centre Demographic Information System (ACDIS) and population-based HIV survey. *International journal of epidemiology*. 2008;37(5):956-62.

VERSION 2 – REVIEW

REVIEWER	Sonak Pastakia AMPATH / Purdue University (Kenya/USA)
REVIEW RETURNED	01-Sep-2018
GENERAL COMMENTS	I thoroughly enjoyed the revised version and would be eager to share the paper with my colleagues working in this domain once published. It represents a valuable contribution to this field because of its large scale assessment, novel methodology and comprehensive evaluation.